# Management of Anastomotic Leakage after Colorectal Resection: Survey among the German CHIR-Net Centers

**DOI:** 10.3390/jcm12154933

**Published:** 2023-07-27

**Authors:** Flavius Șandra-Petrescu, Nuh N. Rahbari, Emrullah Birgin, Konstantinos Kouladouros, Peter Kienle, Christoph Reissfelder, Emmanouil Tzatzarakis, Florian Herrle

**Affiliations:** 1Surgical Department, Medical Faculty Mannheim, University of Heidelberg, Theodor-Kutzer-Ufer 1-3, 68167 Mannheim, Germany; nuh.rahbari@umm.de (N.N.R.); emrullah.birgin@umm.de (E.B.); konstantinos.kouladouros@umm.de (K.K.); christoph.reissfelder@umm.de (C.R.); emmanouil.tzatzarakis@umm.de (E.T.); florian.herrle@umm.de (F.H.); 2Interdisciplinary Endoscopy, Medical Faculty Mannheim, University of Heidelberg, 68167 Mannheim, Germany; 3Surgical Department, Theresien Hospital, 68165 Mannheim, Germany

**Keywords:** anastomotic preservation, rectal resection, rectal cancer, ostomy, intestinal continuity, endoscopic negative pressure therapy

## Abstract

(1) Background: A widely accepted algorithm for the management of colorectal anastomotic leakage (CAL) is difficult to establish. The present study aimed to evaluate the current clinical practice on the management of CAL among the German CHIR-Net centers. (2) Methods: An online survey of 38 questions was prepared using the International Study Group of Rectal Cancer (ISREC) grading score of CAL combined with both patient- and surgery-related factors. All CHIR-Net centers received a link to the online questionary in February 2020. (3) Results: Most of the answering centers (55%) were academic hospitals (41%). Only half of them use the ISREC definition and grading for the management of CAL. A preference towards grade B management (no surgical intervention) of CAL was observed in both young and fit as well as elderly and/or frail patients with deviating ostomy and non-ischemic anastomosis. Elderly and/or frail patients without fecal diversion are generally treated as grade C leakage (surgical intervention). A grade C management of CAL is preferred in case of ischemic bowel, irrespective of the presence of an ostomy. Within grade C management, the intestinal continuity is preserved in a subgroup of patients with non-ischemic bowel, with or without ostomy, or young and fit patients with ischemic bowel under ostomy protection. (4) Conclusions: There is no generally accepted therapy algorithm for CAL management within CHIR-Net Centers in Germany. Further effort should be made to increase the application of the ISREC definition and grading of CAL in clinical practice.

## 1. Introduction

Colorectal anastomotic leakage (CAL) is one of the most feared complications after colorectal surgery, with an incidence of up to 20% [1,2]. CAL was shown to correlate with increased morbidity, poorer functional and oncological outcomes, and quality of life [3,4,5]. Altogether there is an increased clinical and economic burden caused by a leaking colorectal anastomosis [6]. Early diagnosis and contemporary management of CAL are therefore critical in colorectal surgery. However, effective management of CAL remains a clinical challenge. It is influenced by both patient- and surgery-related factors, and a generally accepted algorithm to manage CAL has not been established to date [7,8,9].

In 2010, the International Study Group of Rectal Cancer (ISREC) proposed a definition and grading score of CAL, which considers both leakage features and patient condition [10]. Although the grading has been validated in former studies, its application in the routine clinical work-up remains unclear [8,11,12]. Other studies reported a more patient-tailored algorithm, the decision-making being frequently influenced by the surgeon or hospital experience [13,14,15].

So far, in Germany, the management of CAL in different centers has not been evaluated before. The purpose of the present study was to investigate the current standard therapy of colorectal anastomotic leakage among Surgical Study Network (CHIR-Net) Centers in Germany based on the ISREC grading score of CAL. 

## 2. Materials and Methods

An online questionnaire containing 38 multiple-choice questions on the treatment of CAL after surgery was developed using an online tool provided by ©SurveyMonkey (SurveyMonkey Europe UC, 2nd Floor, 2 Shelbourne Buildings, Shelbourne Road, Dublin, Ireland) (Appendix A–d). Four issues were addressed: the characteristics of the participating centers, the preoperative management for elective colorectal resection, the diagnostic and the management of the CAL. Similarly to other studies, the management of CAL was evaluated in relation to patient condition and age, the leakage localization with respect to the peritoneal cavity and the presence of a deviating ostomy [14]. Additionally, the ISREC definition and the clinically relevant CAL grades B and C, as well as the bowel vascularization and the endoscopic negative pressure therapy (ENPT), complemented the present survey [10,14]. All 14 regional centers, members of the Surgical Study Network (CHIR-Net) by the time the study was developing, and further 26 affiliated centers (*n* = 40), including academic and non-academic centers, received a link to the online questionnaire in February 2020. The survey link was sent to the head of the surgical department, who filled in the questionnaire himself or passed it to the head of the colorectal department. The questionnaire was open for four months. Reminders were sent monthly. The answers were collected anonymously online. Open commentaries related to both survey and CAL management were allowed at the end of the questionary (Appendix A). 

Descriptive statistics are presented as frequencies and percentages for categorical variables and as median and interquartile range (IQR) for continuous variables. 

Definitions: The anastomotic leakage (CAL) and its grade were defined according to the clinical grading classification of the International Study Group of Rectal Cancer (ISREC) [10]: grade A—asymptomatic CAL; grade B—CAL requiring active therapy including antibiotics or interventional drainage but no revision surgery; grade C—CAL requiring re-surgery (laparotomy or laparoscopy). The ISREC grading score reflects both the clinical condition of the patient and the condition at the anastomotic site as it includes parameters such as symptoms and general condition of the patient (A—good; B—mild/moderate discomfort; C—severely impaired), leakage features, characteristics of the drain secretion and laboratory tests [10]. A leaking anastomosis situated caudally from the sacral promontory was referred to as an extraperitoneal leakage. The German categorization of hospitals is based on bed capacity and complexity of provided medical care and includes the following levels of care: tertiary, including academic and non-academic hospitals, and secondary and primary level of care. Following approaches were regarded as trans-anal endoscopic procedures: TEO—trans-anal endoscopic operation; TEM—trans-anal endoscopic microsurgery; TAMIS—trans-anal minimally invasive surgery. The term “percutaneous drainages of a pelvic abscess” refers to the placement of transabdominal and/or trans-gluteal drainages, both computed tomography and ultrasound-guided.

## 3. Results

### 3.1. Characteristics of the Participating Centers (Appendix A)

A total of 22 out of 40 centers (55%) completed the questionnaire. Of these, the majority are tertiary-care academic hospitals. Almost half of the study centers are certified by the German Cancer Society (DKG) as oncological and one-third as colorectal centers. Additionally, almost one-third of the study centers are certified by the German Society of General and Visceral Surgery (DGAV) as colorectal as well as centers for minimally invasive surgery and participate in the national registry of the DGAV for quality management for colon and rectal cancer as well as for diverticulitis (StudoQ). 

Among the study centers, a median of 150 [IQR 100–275] colorectal resections with primary anastomosis were performed per year. Of these, 70 [IQR 50–110] and 40 [IQR 28–65] were resections for colon and rectal cancer, respectively.

### 3.2. Perioperative Management (Appendix A)

Perioperative antegrade mechanical bowel preparation (MBP) is performed in more than two-thirds of the study centers (MBP centers). More than two-thirds of the MBP centers apply it for colorectal resections both with and without protective ostomy, whereas one-third prefer MBP only if a concomitant protective ostomy is formed. Further, MBP is routinely performed within half of the centers by antegrade lavage and concomitant non-absorbable oral antibiotics. Intravenous antibiotics are additionally used in more than one-third of the MBP centers. In two centers, MBP is performed solely by antegrade lavage or in combination with intravenous antibiotics. 

### 3.3. Diagnosis of a Leakage of the Colorectal Anastomosis (Appendix A)

Two-thirds of the study centers reported an average annual incidence of CAL of less than 10%. The most frequent diagnostic method to detect CAL was a CT scan with contrast enema (86.4%) and endoscopy (73%). Only half of the study centers routinely apply the ISREC grading in the management of CAL in clinical practice.

### 3.4. General Aspects of the Management of Colorectal Anastomotic Leakage (Appendix A)

More than two-thirds of the study centers consider that a therapy algorithm would be useful for the management of CAL. Generally, it is preferred to preserve both intra- and extraperitoneal leaking anastomoses (72.7 and 81.8%) in young patients having a good clinical condition (ASA 1–2 and <80 years of age). In elderly and/or frail patients (ASA ≥ 3 and/or ≥80 years of age), only 36.4% of the centers would preserve an intraperitoneal, whereas 59.1% an extraperitoneal anastomosis. If an extraperitoneal anastomosis should be repaired, more than half of the centers (54.6%) prefer the option of a transanal approach, particularly if the anastomosis is located within 6 cm from the anal verge (50 vs. 27.2%). In the case of the latter, conventional surgery is preferred to the endoscopic approach (66.7 vs. 25%). The majority of the centers favored endoscopic negative pressure therapy (ENPT) to manage CAL, particularly if the anastomotic dehiscence was smaller than 50% of the circumference. Also, diverting ileostomy is the preferred approach to protect CAL instead of a protective colostomy (81.8 vs. 9.1%).

### 3.5. Management of Colorectal Anastomotic Leakage Based on ISREC Grading in Combination with the Clinical Condition of the Patient, Presence of Ostomy and Bowel Perfusion

#### 3.5.1. Clinical Condition of the Patient and Good Blood Perfusion at the Anastomotic Site

In the absence of a protective ostomy, half of the centers would treat the leakage in young and fit patients as grade B. The rest of the study centers (40.9%) would treat the same patients as a grade C leakage but would preserve the anastomosis. Meanwhile, if elderly and frail patients without primary deviating ostomy are considered, more than half of the study centers (59.1%) would treat CAL as a grade C leakage. Remarkably, more than two-thirds of these centers (68.2%) prefer to preserve the anastomosis in this case.

If diverting ostomy was primarily formed, all centers prefer to manage CAL as a grade B in both young and fit and elderly and/or frail patients. However, assuming a grade C leakage in the presence of a diverting ileostomy, most of the centers would salvage the anastomosis in young and fit and two-thirds in elderly and/or frail patients Table 1.

#### 3.5.2. Clinical Condition of the Patient and Poor Perfusion of the Anastomosis

Poor perfusion of the anastomosis, as diagnosed by the endoscopy, is considered an indication for surgery in most of the centers (grade C), in both young and fit, and elderly and/or frail patients, irrespective of the presence of protective ileostomy (86.4% without and 77.3% with ostomy). If no ostomy is present, the most applied procedure would be a takedown of the anastomosis and formation of a permanent colostomy in both young and fit and elderly and/or frail patients (59.1 and 72.7%, respectively). If CAL is protected by a primarily formed ileostomy, half of the centers will preserve the anastomosis in young and fit patients, while most of them (81.8%) would still prefer to take it down in elderly and/or frail patients (Table 1).

### 3.6. Methods to Manage a Leaking Colorectal Anastomosis

Figure 1 and Figure 2 depict the methods used among the study centers to manage a CAL with respect to several patient- and anastomosis-related factors. 

ENPT is the most commonly used method to preserve the anastomosis in both young and fit and elderly and/or frail patients in case of good perfusion at the site of the anastomosis and irrespective of the presence of a deviating ostomy. Moreover, in case of good perfusion, one-third of the centers would combine ENPT with stoma formation (grade C) in order to preserve CAL in both patient groups. The most commonly used method to salvage a CAL grade C, which is protected by ostomy, is an anastomotic repair or redo, irrespective of tissue perfusion at the site of the anastomosis in young and fit patients. In elderly and/or frail patients, the anastomotic repair or redo remains the preferred approach (45.5%) to save CAL protected by ostomy only in case of good tissue perfusion. Considering the young and fit patients experiencing CAL in the absence of a deviating ostomy, almost half of the centers would prefer to repair the anastomosis and concomitantly form an ileostomy.

Appendix A depict the participants’ remarks regarding the survey and management of the CAL, respectively. Both of them underline a persistent trend to apply a more patient-tailored management and that a generally accepted algorithm does not exist. However, several important aspects of CAL management were identified, such as the time-point of intervention, the use of prophylactic ENPT, the relevance of post-CAL morbidity and the lack of high-level evidence for CAL management. 

## 4. Discussion

The current study used the ISREC clinical grading of CAL for the first time in order to evaluate CAL management among the members of CHIR-Net in Germany. Despite abundant research on the therapy of colorectal disease, there is still no widespread algorithm for the management of CAL, which is the most feared complication after colorectal surgery. This may partly explain the large variability of the incidence of CAL reported among clinical studies [16]. In the present study, 40 centers, members of CHIR-Net and affiliated centers, were anonymously interviewed via an online survey. The response rate was similar to other studies [14,15]. Remarkably, 77% of the answering study centers agreed that a standardized algorithm will help to better manage a leakage but only half of them use the definition and grading of CAL proposed in 2010 by ISREC [10]. The latter was previously proved to be easy to apply and clinically relevant with impact on the management of CAL [11,12]. Similarly, a recent study using a Delphi round among the Italian surgical societies reached a strong consensus on the standard use of the ISREC definition and grading of CAL [8]. In contrast, the earlier studies of Phytaiakorn et al. and Daams et al. did not use a standard definition and grading when evaluating the management of CAL using a Delphi method among colorectal and radiology experts and a survey among the Dutch surgeons, respectively [13,14]. However, both studies, which were conducted before the ISREC grading was published, show a preference for a patient-tailored therapy and a tendency towards the treatment of CAL by decision-making at the time of diagnosis, depending on center or surgeon experience [13,14]. Similarly, Rink et al., who recently used a Delphi round among German colorectal experts in order to evaluate the impact of the perioperative management and surgical technique on CAL, did not apply the definition or grading of CAL as proposed by ISREC [7]. Furthermore, a recent survey among Dutch and Chinese colorectal surgeons could not identify a standard definition of CAL being commonly used in both countries [9]. This suggests that there is still uncertainty in applying a standardized definition or grading score for the management of CAL.

Management of CAL is complex and should include prevention, diagnosis and treatment pathways. The preoperative antegrade mechanical bowel preparation (MBP) combined with non-absorbable oral antibiotics (ABX) before elective colorectal surgery has been proven to be efficient in reducing the rate of CAL and surgical site infections [17,18,19]. Despite this, our survey, similar to a previous German study, shows that MBP combined with oral ABX is not routinely used (36.4%) [16]. CT scan with contrast enema was used within most of the centers of the current study (86%) as a diagnostic method for patients with suspected CAL. Of these, 73% combined it with endoscopy. These data support a previous consensus on CAL diagnostic methods among colorectal experts in Germany and Italy [7,8]. More than half of the survey centers had an annual overall incidence of CAL of 5.4%, while in almost one-third of the centers, this reached up to 11.9%, similar to other studies [16].

Both patient- and surgery-related factors were used in the present study in combination with the ISREC definition and grading of CAL in order to evaluate the current algorithm of therapy within German CHIR-Net centers [8,9,11,13,14]. It is generally preferred to preserve a leaking anastomosis in young and fit patients, for both intra- and extraperitoneal localization, but also in elderly and/or frail patients in case of extraperitoneal anastomosis. The earlier study by Daams et al. shows a similar but slighter tendency only in young and fit patients [14]. Phitayakorn et al. assessed the management of CAL according to the anastomotic site and patient condition, as well as the presence of a deviating ostomy, and further proposed to generally take down the anastomosis in septic patients [13]. In contrast, the present in accordance with other recent studies, suggests a trend towards preservation of the intestinal continuity in the case of CAL [8,11]. A recent German Delphi study showed, after all, an increased acceptance for saving a leaking anastomosis in the case of stable patients, good bowel vascularization and a drained leak cavity [7]. However, the study did not focus on the management of CAL. Interestingly, both former studies did not consider ENPT as an approach to salvage a leaking anastomosis [13,14]. Meanwhile, endoscopy has been proven to be both an efficient diagnostic and treatment (i.e., ENPT) method within both the upper and lower gastrointestinal tract, which might be an explanation for the above-mentioned trend toward anastomotic preservation [11,20,21,22]. In the current study, ENPT was preferred in the treatment of CAL in 82% of the centers. Moreover, half of the study centers apply ENPT for leakage of up to 50% of the circumference, in contrast to other studies which would treat such a CAL as a grade C leakage [13]. The endoscopic or conventional trans-anal anastomotic repair were important methods used within the study centers to preserve a leaking anastomosis, which is in consensus with the study of Talboom et al. [15]. Altogether, this emphasizes the efforts within the present study centers to salvage a leaking anastomosis and thus to avoid a permanent ostomy, which is in accordance to other studies [15].

Deviating ostomy influenced decision-making in the treatment of a leaking anastomosis in the present survey. Firstly, in the absence of a deviation, slightly more centers regarded CAL as grade B in the case of young and fit patients (grade B vs. C: 50.1 vs. 40.9%). The preference changes in the case of elderly and/or frail patients when more than half of the study centers prefer to perform surgery (grade B vs. C: 31.8 vs. 59.1%). Similar to other studies, poor vascularization was in most of the centers (86.4%), a criterion to manage CAL as grade C in all patients [11,15]. Considering a grade C leakage with good bowel vascularization, all centers would try to salvage the anastomosis and form a protective ostomy in young and fit patients. In contrast, for elderly and/or frail patients, 22.7% of the centers would dissolve the anastomosis, which is still less in comparison to other studies [13]. Similar to other studies, the current participating centers prefer to form a loop ileostomy as fecal diversion [1,15]. Poor blood perfusion at the site of the anastomosis leads to more discontinuity resections, especially for elderly and/or frail patients (59.1 vs. 72.7%). Nevertheless, one-third of the centers would try to salvage the anastomosis while forming a deviating ostomy in young and fit and 18.4% in elderly and/or frail patients. A similar tendency toward the preservation of intestinal continuity was recently observed in an international case-vignette survey [15]. Secondly, if a deviating ostomy was primarily formed, more than half of the centers regarded a leaking anastomosis as a grade B leakage and preferred to apply ENPT in both patient groups (68.2 and 59.1%, respectively). A step up to a grade C management was preferred only in case of ischemic anastomosis. In this case, almost half of the study centers still preferred to preserve bowel continuity by repairing or redoing the anastomosis in young and fit patients. This may be explained due to the fact that less inflammation is expected at the site of the leakage since a protective ostomy is known to reduce the clinical impact of the leakage [23]. Nevertheless, for elderly and/or frail patients, 82% of the centers prefer to take down the anastomosis, probably due to the patient’s general condition, as also reported by other studies [11,14]. Tissue perfusion at the site of the anastomosis was evaluated in this study by endoscopy at the time of the diagnosis of CAL. The use of fluorescence-based diagnostic methods was not assessed in the present survey [24]. Although fluorescence angiography using indocyanine green is increasingly used to assess bowel perfusion prior to performing a colorectal anastomosis, there is still uncertainty regarding its efficiency in reducing the rate of CAL [24,25,26]. However, further studies are necessary to evaluate the value of florescence-based diagnostic methods in the assessment of a leaking anastomosis.

In summary, according to the present survey, a CAL may be managed in the presence of primary ostomy and in case of good vascularization as a grade B leakage. Preservation of the anastomosis should be considered in young and fit patients, independently of its intra- or extraperitoneal localization, and in elderly and/or frail patients in case of extraperitoneal localization. The absence of a deviating ostomy in young and fit patients should not be considered an absolute criterion to manage a leaking anastomosis as grade C. A grade B management should be evaluated in these patients. According to the ISREC definition, patients undergoing grade B management should have a moderately impaired clinical condition. In case of a severely impaired, life-threatening clinical condition, a step up to grade C management should be considered regardless of the former criteria. Grade C management should be considered in all patients in case of ischemic bowel, irrespective of the presence of an ostomy, as well as in elderly and/or frail patients without ostomy protection, irrespective of perfusion status. ENPT may be primarily used to salvage a leaking anastomosis as long as the drainage of a pelvic abscess is guaranteed. In cases of grade C management, Hartmann’s situation may be avoided in favor of anastomotic redo or repair in order to preserve the intestinal continuity and avoid a permanent ostomy; the latter has been previously shown to increase in the last decade [27].

The present study has several limitations. The participating centers are full and affiliated members of the German CHIR-Net and, therefore, not quite representative of all centers performing rectal resection in Germany. Most of the centers are academic hospitals and have expertise in colorectal surgery as certified by DKG and DGAV. However, ENPT might not be available in all participating centers as an anastomosis salvaging method. Therefore, a preference for grade C management might be present. Potential bias may be present due to ambiguity of the used terms and due to the given form of the questions. Therefore, only few options for patient-tailored decision-making are given due to the study design, which, however, is meant to evaluate a standard algorithm of CAL management. Nevertheless, both anastomotic features and patient conditions are included in the ISREC definition and grading of CAL. Therefore, it widely reflects clinical conditions which are needed to choose patient-adapted management of CAL. However, the use of ISREC scores should be expanded in order to be able to compare the results and evaluate whether a generally accepted management algorithm of the CAL can be established. This may further allow, in combination with predictor scores, peri- and intraoperative management pathways or application of the multiprofessional enhanced recovery after surgery (ERAS^®^) treatment pathway to improve the management of CAL [28,29,30]. 

## 5. Conclusions

There is no generally accepted management algorithm of CAL among the Chir-Net centers in Germany. A trend was observed toward the preservation of the bowel continuity by salvaging a leaking anastomosis. Larger international studies are necessary in order to standardize the management of a leaking colorectal anastomosis. Management of CAL according to the ISREC grading score is feasible since both leakage features and patient conditions are included. As, however, the score is not used routinely by a significant number of centers, further efforts are needed to increase the use of the ISREC grading score in clinical practice. 

## Figures and Tables

**Figure 1 jcm-12-04933-f001:**
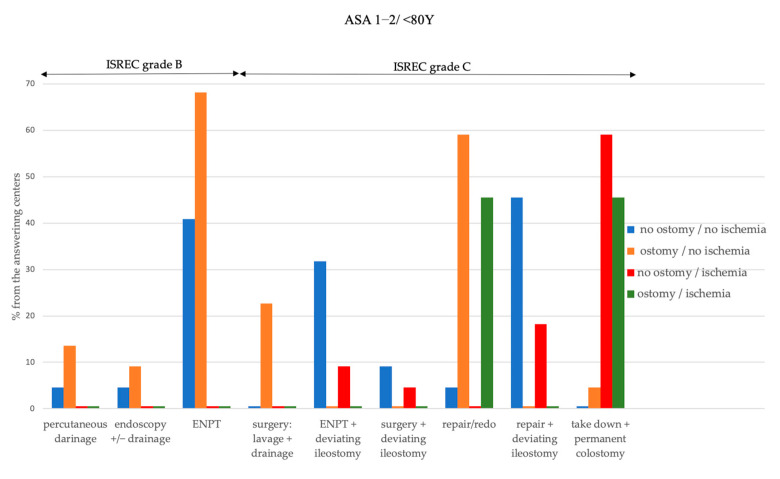
Overview of the methods used among the survey centers to manage a leaking colorectal anastomosis in young and fit patients in relation to the ISREC grading of CAL, presence of a protective ostomy and blood perfusion at the site of the anastomosis.

**Figure 2 jcm-12-04933-f002:**
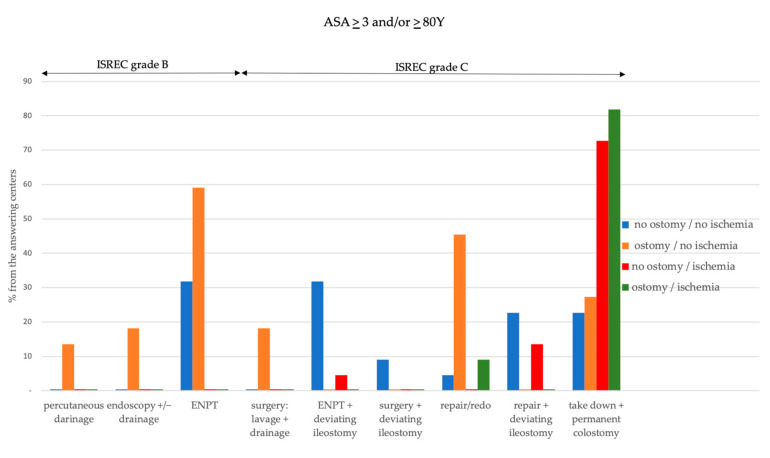
Overview of the methods used among the survey centers to manage a leaking colorectal anastomosis among the study centers in elderly and/ or frail patients in relation to the ISREC grading of CAL, presence of a protective ostomy and blood perfusion at the site of the anastomosis.

**Table 1 jcm-12-04933-t001:** Summary of the preferences of the study centers regarding the management of CAL according to ISREC grading and clinical condition. Grade C leakages were split into two groups: re-surgery with or without preservation of the anastomosis.

ISREC Grade	ASA 1–2; <80Y	Ostomy	Ischemia	Extraperitoneal
B (antibiotics, percutaneous or transanal drainage, transanal lavage)	+ *	−	−	+/−
	+ **	+	−	+/−
	−	+	−	+
C (re-surgery: laparotomy or laparoscopy)				
preservation of the anastomosis	+ */−	−	−	+/−
	+ **/−	+	−	+/−
	+	+	+	+
anastomosis take down	+/−	−	+	+/−
	−	+	+	+/−

ASA American Society of Anesthesiologists, CAL colorectal anastomotic leakage, ISREC International Study Group of Rectal Cancer, Y years old. + condition is available, − condition is missing, * and ** group of patients which would be treated either as grade B or C.

## Data Availability

Data presented in this study are contained within this article and Appendix A.

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
