# Peer review of "Management of Anastomotic Leakage after Colorectal Resection: Survey among the German CHIR-Net Centers"

_jcm, 2023, doi:10.3390/jcm12154933_

Round 1

Reviewer 1 Report

Dear Authors, 

I would like to congratulate you for conducting the study regarding the management of anastomotic leakage after colorectal resection among a set of specialised centres in your country. 

I find your manuscript to be quite well written, significant for the surgical community and, overall, a representative portrait of the current state of patient management in the field.

The only part that I consider to need improvements is the two figures you have included: 1. please make sure spelling mistakes on the x-axis are corrected; 2. please use the y-axis  only for the interval you need (Fig 1 could have stopped at 70); 3. please include the title and if it is expressed as a percentage or number of responders on the y-axis; 4. please remove the border of the graph; 5. please increase the font size of the labels; 6. please write a more comprehensive caption for the figure; 7. Figure captions are to be placed below, not above the graphs. 

Best regards, 

Your Reviewer

Author Response

  1. please make sure spelling mistakes on the x-axis are corrected;

Thank you very much for reviewing the manuscript and for your suggestions. The text was changed accordingly.

  1. please use the y-axis  only for the interval you need (Fig 1 could have stopped at 70);

The figure was changed accordingly

  1. please include the title and if it is expressed as a percentage or number of responders on the y-axis;

A text was added accordingly

  1. please remove the border of the graph;

The figures were changed accordingly

  1. please increase the font size of the labels;

The figures were changed accordingly

  1. please write a more comprehensive caption for the figure;

The text was changed accordingly

  1. Figure captions are to be placed below, not above the graphs.

The figures were changed accordingly

Reviewer 2 Report

Overall this is a well-written survey study of the management of colorectal leak in German CHIR-Net Centers. The introduction is clear and well referenced. When reviewing the results of the paper it would be helpful to know what percentage of patients had emergency surgery and neoadjuvant therapy. A clearer delineation between colon and rectal anastomoses and more precise location of the anastomoses would be useful to better characterize leak management. The header for figures one and two regarding perfusion are unclear as there are separate bars for ischemia. The discussion flow well, although claims between lines 284-299 may be overly bold especially considering the survey nature of this study and in the absence of outcome data. There are certainly applicability limitations as mentioned in the discussion with most centers participating in the survey being academic centers with colorectal expertise. 

Author Response

Overall this is a well-written survey study of the management of colorectal leak in German CHIR-Net Centers. The introduction is clear and well referenced. When reviewing the results of the paper it would be helpful to know what percentage of patients had emergency surgery and neoadjuvant therapy.

- Thank you very much for reviewing the manuscript and for the suggestions you made. The questionnaire was designed to investigate the management of CAL thus, no information about the neoadjuvant or adjuvant therapy was collected. This aspect is definitely of high importance when analyzing the incidence of CAL and it is included in a future project.

A clearer delineation between colon and rectal anastomoses and more precise location of the anastomoses would be useful to better characterize leak management.

- The survey refers only to elective resections and includes only colorectal anastomosis (i.e. descend-rectostomy). The cases presented in the questionnaire are only rectal resections. However, the distance of the anastomosis to the anal verge was addressed in the question 18a. An additional text was added in the section Material and Methods.

The header for figures one and two regarding perfusion are unclear as there are separate bars for ischemia.

- The header of the figures was changed.

 The discussion flow well, although claims between lines 284-299 may be overly bold especially considering the survey nature of this study and in the absence of outcome data. There are certainly applicability limitations as mentioned in the discussion with most centers participating in the survey being academic centers with colorectal expertise. 

- The discussion section was changed accordingly.

Reviewer 3 Report

Thank you for submitting this interesting survey study, showing there is still a heterogeneity in the management of colorectal anastomotic leaks. I totally agree with the statement that further effort should be made to improve an uniform accepted definition of CAL and grading system! Great to see this was investigated in Germany among so many centers.

I have a few questions/comments on the manuscript as I believe it can be described more clearly.  

In general

Provide an explanation of the ISREC classification; or in the manuscript or in the supplementary. As you are talking about grade B and C often, it's important the reader knows well what this means. As you stated in the manuscript, only half of the centers do use this classification, so it's likely for readers to not know this in detail. 

Were there reminders send? As only 22 out of 40 replied.
How did the filling in of the survey worked? Did surgeons from every center had to discuss the survey and complete one file? Or did the head of the surgery department do it him/herself? Was this person always a colorectal expert? Can influence outcomes. 

A nice extra question would've been: if centers do not use ISREC, how do they define a leak? suggestion. 

3.2 Perioperative management

How is MBP done? oral/enema?
You mention that the use of oral antibiotics has been investigated. Did you also have a look of what type of antibiotics? research is showing a higher interest in selective decontamination compared to broad spectrum AB so it might be interesting to make a subdivision in this, as both intervention show different outcomes regarding leakage rates. 
Can you add timeframe of which the perioperative actions are undertaken? one day prior surgery? more days? may be different across the centers.

3.3 Diagnosis

In what timeframe do the centers consider an CAL? This is not integrated in the ISREC classification, while it is an important topic. 

3.4 General aspects

In case of surgery, do they prefer minimal invasive approach or open surgery? it's stated that in case of anastomosis within 6cm from the anal verge, most surgeons prefer endoscopic approaches. But it's not stated when laparoscopy or laparotomy approaches are considered. ISREC grade C recommends re-laparotomy, but in more stable patients minimal invasive approaches are also considered nowadays. Important to mention how redo-surgery is performed. 

3.5 Management

Table 1: Good that you combine ISREC classification with ASA, ostomy, ischemia and location information. In the table it is not very clear why there are different treatment options? Are these the ones suggested by the experts? Also what type of B and C ISREC? B is antibiotics or drainage, so not clear what difference is made here in the 3 options. --> more explanation necessary, it's not reflecting well the questions and answers as written in the supplementary. 

After reading the other sections: I feel the table is not well constructed/reflecting what you want to say. Only after reading the sections afterwards, I understood what the table was meaning and it should be clear on itself. Please re-look. 

3.6 methods to manage CAL

fig 1 and 2:
- in the questionnaire you are only asking questions about rectal resections. this should be stated, as there is now 'colorectal' in both figures titles.  --> what about other interventions?

- clarify y-axis. Percentage of? Cause the numbers are reflecting the question for a specific case, not for the intervention. I'm not sure this it the right type of graph to show these results.
- 'and perfusion' --> why is it in the title? as the legendary is showing perfusion state and ostomy state. 
- 'take dowm' = take down
- 'Drainage' = percut. drainage, confusing as we can also do surgical drainage as stated in your survey options.
- What's the difference between surgery + ostomy / repair + ostomy? clarify.
- Take down alone: shouldn't there be 'take down + ostomy'? Cause both fig 1 and 2 show patient without and with ostomy are treated like this. But what's the difference with surgery + ostomy then and take down + ostomy? Do you mean surgical drainage in the first example too?
--> would re-look at how the questions were asked an rename the x-axis, now it's confusing and not reflecting the answers given in the survey. I assume it should be more like?:
Percut. drainage - Endo/drainage - ENPT - Surgical drainage - ENPT + ostomy - surgical drainage + ostomy - surgical repair/redo - surgical repair/redo + ostomy - surgical take down + ostomy
- There is no difference made in ileostomy or colostomy, while the survey is making a difference in these. 

In general: how was perfusion assessed? Just by surgical look or did some use fluorescence? 

4. Discussion

Clarify response rate in the discussion section. as you state 77% of the study centers agreed that standardized alghoritm will help, it looks like 77% of the 40 contacted centers. 

As the perfusion status is influencing decision making, shouldn't we improve perfusion assessment then? Is fluorescence used in centers? Think it would be a nice not in the discussion. 

Limitations: ... "therefore, preference for a grad C management..." --> grade

The discussion is long, if possible a little shorter would be more feasible. Often repeating the results

None, some small mistakes (mentioned in the comments). 

Author Response

Thank you for submitting this interesting survey study, showing there is still a heterogeneity in the management of colorectal anastomotic leaks. I totally agree with the statement that further effort should be made to improve an uniform accepted definition of CAL and grading system! Great to see this was investigated in Germany among so many centers.

I have a few questions/comments on the manuscript as I believe it can be described more clearly.  

In general

Provide an explanation of the ISREC classification; or in the manuscript or in the supplementary. As you are talking about grade B and C often, it's important the reader knows well what this means. As you stated in the manuscript, only half of the centers do use this classification, so it's likely for readers to not know this in detail. 

- Thank you for reviewing the manuscript and your suggestions. The text in the Methods Section/Definition was changed accordingly.

Were there reminders send? As only 22 out of 40 replied.
How did the filling in of the survey worked? Did surgeons from every center had to discuss the survey and complete one file? Or did the head of the surgery department do it him/herself? Was this person always a colorectal expert? Can influence outcomes. 

- Thank you for the suggestion. The text in the Material and Methods Section was changed accordingly.

A nice extra question would've been: if centers do not use ISREC, how do they define a leak? suggestion. 

- Although this question would have improved the quality of the questionnaire, it was not directly addressed. Therefore, now answer can be given at this time. We tried to keep the questionnaire as short as possible in order to achieve more compliance and a high rate of answers. Thank you very much for the suggestion which we will use in a future project.

3.2 Perioperative management

How is MBP done? oral/enema?

- MBP was performed orally. This is mentioned in the section Perioperative Management as “antegrade lavage”. A supplementary text was added here but also in the table S1b in order to underline the idea. Thank you very much for the suggestion.

You mention that the use of oral antibiotics has been investigated. Did you also have a look of what type of antibiotics? research is showing a higher interest in selective decontamination compared to broad spectrum AB so it might be interesting to make a subdivision in this, as both intervention show different outcomes regarding leakage rates. 

- Thank you very much. As mentioned above we tried to keep the questionnaire as short as possible. No further investigation regarding the type of the antibiotics was done. However, there were used non-absorbable oral antibiotics. We mentioned now in the text.

Can you add timeframe of which the perioperative actions are undertaken? one day prior surgery? more days? may be different across the centers.

- Thank you very much. We perform MBP in our clinic one day before surgery but this question was not addressed in the survey.

3.3 Diagnosis

In what timeframe do the centers consider an CAL? This is not integrated in the ISREC classification, while it is an important topic. 

- Thank you very much. A timeframe for the diagnosis of the CAL was not established or investigated by the present study. However, this is a shortcoming of the ISREC grading (which I hope it will be soon upgraded) since the acute and chronic CAL but also the anastomotic fistula are not especially defined. According to the ISREC CAL definition a clinically relevant leakage (grade B or C) is mostly diagnosed by CT scan and confirmed by endoscopy which are triggered by patient condition. Both methods are mostly used within our study centers.

3.4 General aspects

In case of surgery, do they prefer minimal invasive approach or open surgery? it's stated that in case of anastomosis within 6cm from the anal verge, most surgeons prefer endoscopic approaches. But it's not stated when laparoscopy or laparotomy approaches are considered. ISREC grade C recommends re-laparotomy, but in more stable patients minimal invasive approaches are also considered nowadays. Important to mention how redo-surgery is performed. 

- Thank you very much. Indeed, the ISREC grading mentions only the re-laparotomy as a re-intervention method. We introduced laparoscopy as an option for re-surgery in order to avoid an exclusion of the laparoscopic procedures due to misunderstanding of the grading score. Therefore, both open and laparoscopic procedures are covered by the “re-surgery” which is used within the survey. A differentiation between the mentioned procedures was not considered when preparing the questionnaire. We better defined this within the section Definitions. It is also explained at the bottom of the supplementary tables.

3.5 Management

Table 1: Good that you combine ISREC classification with ASA, ostomy, ischemia and location information. In the table it is not very clear why there are different treatment options? Are these the ones suggested by the experts? Also what type of B and C ISREC? B is antibiotics or drainage, so not clear what difference is made here in the 3 options. --> more explanation necessary, it's not reflecting well the questions and answers as written in the supplementary. 

After reading the other sections: I feel the table is not well constructed/reflecting what you want to say. Only after reading the sections afterwards, I understood what the table was meaning and it should be clear on itself. Please re-look. 

- Thank you very much. The text was changed accordingly.

3.6 methods to manage CAL

fig 1 and 2:
- in the questionnaire you are only asking questions about rectal resections. this should be stated, as there is now 'colorectal' in both figures titles.  --> what about other interventions?

- within the figures it is mentioned the art of the anastomosis i.e. “colorectal anastomosis” meanwhile the text refers to the resection i.e. “rectal resection” which was the solely procedures included in the survey.

- clarify y-axis. Percentage of? Cause the numbers are reflecting the question for a specific case, not for the intervention. I'm not sure this it the right type of graph to show these results.

- A title of the Y-axis was added. Several diagrams were tested since it was difficult to represent different procedures/parameters within one image. We found the present diagrams to reflect at the best the entire management of CAL. The numbers represent the percentage of the centers (from the total answering centers) that apply a specific management of CAL in a given case/situation.

- 'and perfusion' --> why is it in the title? as the legendary is showing perfusion state and ostomy state. 

- The text was changed accordingly

- 'take dowm' = take down:

- The text was changed accordingly.

- 'Drainage' = percut. drainage, confusing as we can also do surgical drainage as stated in your survey options.

- The text was changed accordingly

- What's the difference between surgery + ostomy / repair + ostomy? clarify.
- Take down alone: shouldn't there be 'take down + ostomy'? Cause both fig 1 and 2 show patient without and with ostomy are treated like this. But what's the difference with surgery + ostomy then and take down + ostomy? Do you mean surgical drainage in the first example too?
--> would re-look at how the questions were asked an rename the x-axis, now it's confusing and not reflecting the answers given in the survey. I assume it should be more like?:
Percut. drainage - Endo/drainage - ENPT - Surgical drainage - ENPT + ostomy - surgical drainage + ostomy - surgical repair/redo - surgical repair/redo + ostomy - surgical take down + ostomy
- There is no difference made in ileostomy or colostomy, while the survey is making a difference in these. 

- Thank you very much for the suggestions. The figures were changed accordingly.

In general: how was perfusion assessed? Just by surgical look or did some use fluorescence? 

- Generally, the bowel perfusion was assessed by endoscopy at the diagnosis of the CAL. Whether an additional assessment of the bowel perfusion at the time of re-surgery (grade C) was not investigated by the survey. This point is extremely important and will be considered in the next planed project.

  1. Discussion

Clarify response rate in the discussion section. as you state 77% of the study centers agreed that standardized alghoritm will help, it looks like 77% of the 40 contacted centers. 

- Thank you very much. We change it accordingly (line 216).

As the perfusion status is influencing decision making, shouldn't we improve perfusion assessment then? Is fluorescence used in centers? Think it would be a nice not in the discussion. 

- many thanks for the suggestion. A text was added accordingly (line 298- 310).

Limitations: ... "therefore, preference for a grad C management..." --> grade

- Thank you very much. We change it accordingly.

The discussion is long, if possible a little shorter would be more feasible. Often repeating the results.

- The text was partially adapted but we think that a significant shortage will lead to a loss of the information that we want to transmit.

Round 2

Reviewer 3 Report

I completely happy with the revisions and would like to accept the revised article

I completely happy with the revisions and would like to accept the revised article